# Knowledge and motivations of training in peer review: An international cross-sectional survey

Jessie V. Willis[1,2]*, Janina Ramos[1,3], Kelly D. Cobey[4,5], Jeremy Y. Ng[1], Hassan Khan[1,5], Marc A. Albert[6,7], Mohsen Alayche[1,2], David Moher[1,5]

1 Centre for Journalology, Clinical Epidemiology Program, Ottawa Hospital Research Institute, Ottawa, Canada, 2 Department of Medicine, Faculty of Medicine, University of Ottawa, Ottawa, Canada, 3 Department of Biology, Faculty of Science, University of Ottawa, Ottawa, Canada, 4 University of Ottawa Heart Institute, Ottawa, Canada, 5 School of Epidemiology and Public Health, Faculty of Medicine, University of Ottawa, Ottawa, Canada, 6 Telfer School of Management, University of Ottawa, Ottawa, Canada, 7 Blueprint Translational Research Group, Ottawa Hospital Research Institute, Ottawa, Canada

* jessievwillis@gmail.com

**Data Availability Statement:** The study protocol was registered to the Open Science Framework (OSF) prior to data analysis (https://osf.io/wgxc2/) 23. Text for this manuscript was drawn directly in

## Abstract

### Background

Despite having a crucial role in scholarly publishing, peer reviewers do not typically require any training. The purpose of this study was to conduct an international survey on the current perceptions and motivations of researchers regarding peer review training.

### Methods

A cross-sectional online survey was conducted of biomedical researchers. A total of 2000 corresponding authors from 100 randomly selected medical journals were invited via email. Quantitative items were reported using frequencies and percentages or means and SE, as appropriate. A thematic content analysis was conducted for qualitative items in which two researchers independently assigned codes to the responses for each written-text question, and subsequently grouped the codes into themes. A descriptive definition of each category was then created and unique themes–as well as the number and frequency of codes within each theme–were reported.

### Results

A total of 186 participants completed the survey of which 14 were excluded. The majority of participants indicated they were men (n = 97 of 170, 57.1%), independent researchers (n = 108 of 172, 62.8%), and primarily affiliated with an academic organization (n = 103 of 170, 62.8%). A total of 144 of 171 participants (84.2%) indicated they had never received formal training in peer review. Most participants (n = 128, 75.7%) agreed–of which 41 (32.0%) agreed strongly–that peer reviewers should receive formal training in peer review prior to acting as a peer reviewer. The most preferred training formats were online courses, online lectures, and online modules. Most respondents (n = 111 of 147, 75.5%) stated that difficulty finding and/or accessing training was a barrier to completing training in peer review.

reference to the registered protocol on OSF. Anonymous study data and any analytical code was shared publicly using the OSF and study findings were reported in a preprint and open access publication.

**Funding:** The author(s) received no specific funding for this work.

**Competing interests:** The authors have declared that no competing interests exist.

## Conclusion

Despite being desired, most biomedical researchers have not received formal training in peer review and indicated that training was difficult to access or not available.

## Introduction

Peer review is the predominant quality control measure for scientific publishing regardless of country or discipline [1–3]. Peer review refers to the process by which "peers" are selected to assess the validity and quality of submitted manuscripts for publication [4]. Responsibilities of peer reviewers typically include providing constructive feedback to the authors of the manuscript and sometimes recommendations to journal editors [5, 6].

Despite its foothold in scholarly publishing, peer review is not a standardized process and lacks uniform guidelines [7–10]. Different scholarly publishers have different requirements and responsibilities for their peer reviewers and peer review data is not always made public [11]. Some publishers provide guidelines and training for their peer review process; however, a 2012 study found that only 35% of selected journals provided online instructions for their peer reviewers [12, 13].

It is therefore understandable that many potential peer reviewers feel inadequately trained to peer review. This is especially true for early career researchers; a recent survey showed that 60% of those under 36 years of age felt there is a lack of guidance on how to review papers [14]. Additional studies have shown that training is highly desired by academics [15–17]. In a 2018 survey by Publons, 88% of survey respondents felt training would have a positive impact on the efficacy of peer review. Despite this, 39% of respondents had never received training and 35.8% had self-trained by reading academic literature. Most respondents believed that training should be provided by scholarly publishers or journals and 45.5% believe that it should be a practical online course [18].

Unfortunately, the effectiveness of peer review training has been studied only via small-scale studies on non-online methods (e.g., workshops) with limited evidence of any benefit [19–22]. Our group was unable to identify any randomized-controlled trials regarding how the electronic delivery of peer review guidelines has impacted the knowledge of potential peer reviewers.

In the present study we conducted a large-scale, online survey to provide an up-to-date perspective of international biomedical researchers' views on peer review training. We focused on biomedical researchers as this is our content area and the needs and perspectives of researchers related to peer review may differ by discipline.

## Methods

### Transparency statement

Ethics approval was obtained from the Ottawa Health Science Network Research Ethics Board (OHSN-REB Protocol Number 20220237-01H). Participants were provided with a consent form prior to entering the survey and consent was presumed if they completed the survey. The study protocol was registered to the Open Science Framework (OSF) prior to data analysis (https://osf.io/wgxc2/) [23]. Text for this manuscript was drawn directly in reference to the registered protocol on OSF. Anonymous study data and any analytical code was shared

publicly using the OSF and study findings were reported in a preprint and open access publication.

## Study design

We conducted a cross-sectional online survey of biomedical researchers. The CHERRIES reporting guidelines were used to inform the reporting of our findings [24].

**Participant sampling framework.** We identified a random sample of international biomedical researchers who are actively publishing in peer-reviewed medical journals. We used the Scopus source list to randomly select 100 biomedical journals. The Scopus list was restricted to those journals with an All Science Journal Classification (ASJC) code of 'Medicine' and those that specified the journal was 'active' at the time of searching (November 2021). We excluded journals that indicated that they only published articles in a language other than English. Using the RAND function in Excel, we then randomly selected 100 journals from this list. Subsequently, we visited each of the randomly selected journal websites and extracted the corresponding authors from the last 20 published research articles. Corresponding author email extraction was completed on December 9, 2021. In instances where the journal was not open access and we did not have access via our academic institution, we replaced the journal with another randomly selected journal. We also replaced any journals which had non-functioning links. A total of 26 journals were replaced for either not being in English (n = 7), not being active after 2020 (n = 8), broken link (n = 4), not open access (n = 3), or no emails for corresponding authors listed (n = 4). We have used this broad approach to sampling successfully in previous research [25]. This approach enabled us to identify a population of 2000 randomly selected researchers to invite to our survey.

**Survey.** The survey was purposefully built for this study and was administered using SurveyMonkey software (https://www.surveymonkey.ca/r/7B2JYR6). This was a closed survey; thus, only available to invited participants via our sampling framework. We emailed the identified sample population a recruitment script with a link to the survey. Participation in the survey was voluntary and all data was completely anonymized. The survey was sent on May 23, 2022. We sent all participants a reminder email to complete our survey after one week (May 30, 2022) and two weeks (June 6, 2022), respectively, from the original invitation. The survey was closed after three weeks (June 13, 2022).

The survey contained 37 questions: 1–10 were demographic questions about the participant, 11–15 were regarding level of experience with peer review, 16–23 were opinion-based questions about peer review training, 24–33 were for respondents who have experience running peer review from a journal perspective, and 34–37 were open-ended questions with comment boxes. 33 of the questions were quantitative while four were qualitative. The survey questions were presented in blocks based on content and question type. The survey used adaptive questioning where certain questions appeared based on the participants' previous responses. The full list of survey questions can be found in S1 File.

The survey was created in SurveyMonkey by two authors (JVW, JR). All survey questions were reviewed and piloted by four researchers (HK, JYN, KDC, DM) and two invited researchers outside of the author list. The average time to complete the survey was estimated to be 15 minutes by pilot testers. All questions were optional and could be skipped. We offered participants the option to report their email to be entered into a draw to win one of three $100 Amazon Gift Cards. Email addresses were stored separately from the study data.

## Data analysis

We used SPSS Statistics and Microsoft Excel for data analysis. We reported the overall response rate based on the number of individuals that completed our survey from the sample identified, as well as the survey completion rate (i.e., the number of people who viewed our survey that completed it). We excluded participants from data analysis if they did not complete 80% or more of the survey. We reported quantitative items using frequencies and percentages or means and SE, as appropriate. For qualitative items, we conducted a thematic content analysis of responses in Excel. For this, two researchers (JR, MAA) independently assigned codes to the responses for each written-text question. Codes were then discussed and iteratively updated until there was consensus among the two researchers that best reflected the data. Following this, individual codes were independently grouped into themes by the two reviewers and finalized by consensus. We then created a descriptive definition of each category. We reported the number of unique themes and the number and frequency of codes within each theme.

# Results

## Protocol amendments

Survey roll-out was changed from four weeks to three weeks due to time constraints. Minor revisions were made to the survey questions, recruitment and reminder emails and consent form.

## Participants

**Demographics.**   A total of 186 participants completed the survey of the 2000 researchers invited (9.3%). There were 107 (5.4%) instances where the email was unable to be sent and 32 (1.6%) instances where the participant indicated (including auto-replies) an inability to be reached/participate. As these accounted for less than 10% of invited participants, no changes were made to the recruitment strategy. A flowchart detailing these instances can be found in S1 Table.

The average completion rate was 92% and it took, on average, 13 minutes to complete the survey. There were 14 responses that were excluded based on having less than 80% questions answered, thus the final included number was 172. A total of 97 of 170 respondents (57.1%) identified as men. The survey received responses from 48 different countries with the greatest representation from United States (n = 41, 24.0%), United Kingdom (n = 13, 7.6%) and India (n = 13, 7.6%). The majority of respondents identified as independent researchers defined as assistant, associate, or full professors (n = 108 of 172, 62.8%), were primarily affiliated with an academic organization (n = 103 of 170, 62.8%), and had published more than 21 peer-reviewed articles (n = 106 of 172, 61.6%). Full demographics are described in Table 1.

**Experience with peer review and peer review training.**   In total, 144 of 171 participants (84.2%) have never received formal training in peer review. The majority answered that their primary institution did not offer peer review training (n = 108, 63.2%) or otherwise did not know of any training offered (n = 48, 28.1%). For those (n = 26) that had received peer review training, the most common training formats were in-person lectures (n = 12, 44.4%), online lectures (n = 10, 37.0%), or online courses of at least 6 sessions (n = 10, 37.0%). Most of the training received was provided by an academic organization (n = 18, 66.7%). Less than half (40.7%) of participants indicated the training was completed over 5 years ago.

For their first-time performing peer review, 88 of 166 (53.0%) participants felt either very unprepared (10.8%), unprepared (24.1%), or slightly unprepared (18.1%). Highlighted

**Table 1. Demographic data.**

| | | Frequency | Percent |
|---|---|---|---|
| Age | 18–24 | 1 | .6 |
| | 25–34 | 30 | 17.4 |
| | 35–44 | 60 | 34.9 |
| | 45–54 | 35 | 20.3 |
| | 55–64 | 26 | 15.1 |
| | 65+ | 19 | 11.0 |
| | Total | 171 | 99.4 |
| Gender | Man | 97 | 56.4 |
| | Woman | 73 | 42.4 |
| | Total | 170 | 98.8 |
| Occupation and/or Position | Other (please specify) | 22 | 12.8 |
| | Master's student | 10 | 5.8 |
| | PhD student | 12 | 7.0 |
| | Post-doctoral fellow | 14 | 8.1 |
| | Independent researcher (e.g., assistant/associate/full professor) | 108 | 62.8 |
| | Research support staff (e.g., research assistant, research coordinator) | 6 | 3.5 |
| | Total | 172 | 100.0 |
| Primary research interest | Other (please specify) | 58 | 33.7 |
| | Clinical | 82 | 47.7 |
| | Pre-clinical ("Basic science") | 30 | 17.4 |
| | Total | 170 | 98.8 |
| Institution | Other (please specify) | 9 | 5.2 |
| | University/college | 103 | 59.9 |
| | Research institute | 4 | 2.3 |
| | Healthcare institution (e.g., medical centre, hospital) | 42 | 24.4 |
| | Private sector (e.g., pharmaceutical company) | 4 | 2.3 |
| | Not-for-profit | 1 | .6 |
| | Government organization | 7 | 4.1 |
| | Total | 170 | 98.8 |
| Scholarly publishing experience | < 1 year | 1 | .6 |
| | 1–5 years | 38 | 22.1 |
| | 6–10 years | 44 | 25.6 |
| | 11–15 years | 29 | 16.9 |
| | 16–20 years | 13 | 7.6 |
| | 21+ years | 47 | 27.3 |
| | Total | 172 | 100.0 |
| Number of peer reviewed articles published to date | < 2 | 5 | 2.9 |
| | 3–5 | 16 | 9.3 |
| | 6–10 | 22 | 12.8 |
| | 11–20 | 23 | 13.4 |
| | 21–50 | 36 | 20.9 |
| | 51+ | 70 | 40.7 |
| | Total | 172 | 100.0 |

responses about peer review and peer review training are shown in Table 2. A complete table of responses can be found in S2 Table.

**Table 2. Experience with peer review and peer review training.**

| | | Frequency | Percent |
|---|---|---|---|
| How many articles have your peer reviewed in the last 12 months? | 0 | 7 | 4.1 |
| | 1–3 | 41 | 23.8 |
| | 4–6 | 38 | 22.1 |
| | 6–10 | 23 | 13.4 |
| | >10 | 58 | 33.7 |
| | I have never been a peer reviewer | 4 | 2.3 |
| | Total | 171 | 99.4 |
| For how many years have you been active as a manuscript peer reviewer? | < 1 year | 11 | 6.4 |
| | 1–5 years | 59 | 34.3 |
| | 6–10 years | 43 | 25.0 |
| | 11–15 years | 15 | 8.7 |
| | 16–20 years | 13 | 7.6 |
| | 21 + years | 28 | 16.3 |
| | Total | 169 | 98.3 |
| Have you completed any formal training in peer review? | Yes | 26 | 15.1 |
| | No | 144 | 83.7 |
| | Unsure | 1 | .6 |
| | Total | 171 | 99.4 |
| Does the primary institution you are affiliated with offer formal training in peer review? | Yes, and I have completed it | 10 | 5.8 |
| | Yes, but I have not completed it | 5 | 2.9 |
| | No | 108 | 62.8 |
| | Unsure/don't know | 48 | 27.9 |
| | Total | 171 | 99.4 |
| Type of formal peer review training completed | | Frequency | Percent |
| | Online lecture | 10 | 16.4 |
| | Online course (at least 6 sessions) | 10 | 16.4 |
| | In-person lecture | 12 | 19.7 |
| | In-person half day workshop | 2 | 3.3 |
| | In-person full day workshop | 7 | 11.5 |
| | Shadowing a mentor/ghost-writing | 4 | 6.6 |
| | Self-selected reading material | 7 | 11.5 |
| | Online resource/module | 8 | 13.1 |
| | Other | 1 | 1.6 |
| | Total | 61 | 100.0 |
| Who provided peer review training that was completed | Journal | 4 | 12.1 |
| | Publisher (of multiple journals | 6 | 18.2 |
| | University/college | 18 | 54.5 |
| | Private company | 2 | 6.1 |
| | Unsure/don't know | 2 | 6.1 |
| | Other | 1 | 3.0 |
| | Total | 33 | 100.0 |

*(Continued)*

**Table 2.** (Continued)

| | | Frequency | Percent |
|---|---|---|---|
| Peer review training provided by institution | Online lecture | 6 | 20.7 |
| | Online course (at least 6 sessions) | 2 | 6.9 |
| | In-person lecture | 3 | 10.3 |
| | Half day workshop | 3 | 10.3 |
| | Full day workshop | 5 | 17.2 |
| | Shadowing a mentor/ghost-writing | 2 | 6.9 |
| | Self-selected reading material | 2 | 6.9 |
| | Online resource/modules | 4 | 13.8 |
| | Unsure/don't know | 1 | 3.4 |
| | Other | 1 | 3.4 |
| | Total | 29 | 100.0 |

## Opinion-based questions

**General statements on peer review and peer review training.** Participants rated their agreement with statements related to peer review and peer review training on a 7-point scale from strongly disagree to strongly agree. A graph of the responses is depicted in Fig 1.

Notable findings included that 148 respondents (86.5%) either strongly agreed or agreed that peer review is important for ensuring the quality and integrity of scholarly communication. One hundred and sixteen (69.5%) strongly agreed or agreed that their experience as a peer reviewer had been positive. Seventy-six (45.2%) strongly agreed or agreed that there is a lack of knowledge and understanding for how to properly conduct peer review. Ninety-nine (58.9%) strongly agreed or agreed that peer reviewers should receive formal training in peer review prior to acting as a peer reviewer for journals. Eighty-six (50.9%) strongly disagreed or disagreed that there were appropriate incentives in place to motivate them to engage in peer review.

## Desired training topics, organizations and formats

These questions required participants to rank their responses in order from most to least preferred. Based on average rank position, a score was given to each response (max score based on number of ranked items). A higher score corresponds to being ranked more highly/preferrable. A graph of the response scores can be found in Fig 2.

The topics that participants were most interested in were appraisal of study design and methodology, appraisal of the research question, and appraisal of statistics. The most desired training formats were all online, including online courses (at least 6 sessions), and online lectures. Academic institutions and scholarly publishers/journals were both similarly ranked as the preferred organization to develop peer review training. Scholarly publishers were additionally ranked as the preferred funders.

**Journal-specific questions.** Participants were only able to answer these questions if they indicated they worked or volunteered for a journal that publishes peer reviewed articles. There were 80 respondents that were included in this section.

In total, 55 of 80 participants (68.8%) indicated that the journal they were affiliated with did not explicitly require peer review training for their reviewers. Eight (10.0%) indicated that it was required and provided by the journal internally, while two (2.5%) indicated that it was required but externally delivered. The most common format of training required was either an

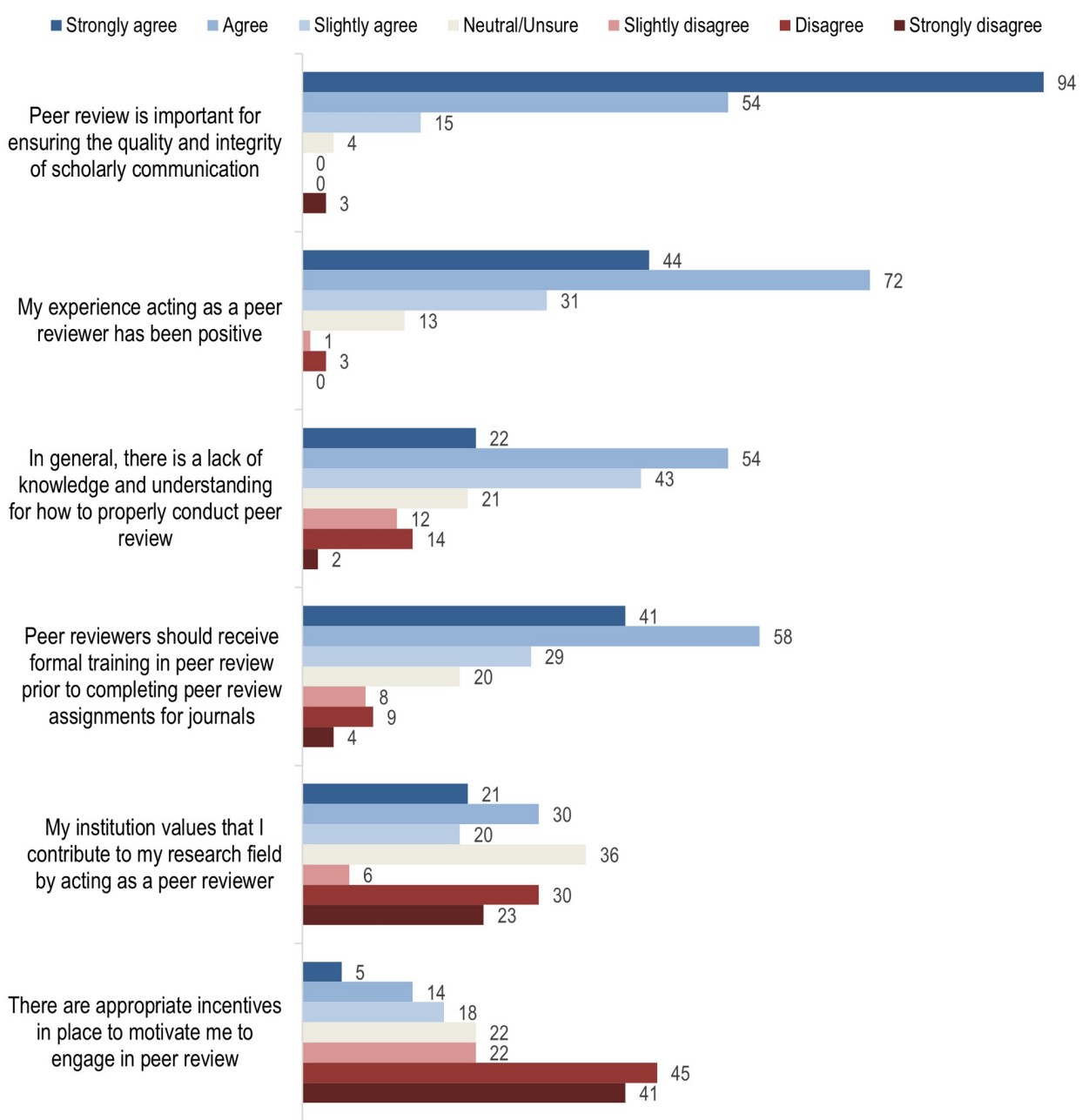

**Fig 1. Participant agreement with statements based on overall experiences with peer review in the last 12 months.**

online course and/or lecture. Required training length was variable from 1–5 hours to 20 + hours.

Only 10 of 80 (10.0%) of participants indicated that the journal assessed peer review reports of new reviewers; however, the majority (n = 51, 63.8%) indicated they were unsure or did not know. Twenty-one (26.6%) provided reporting guidelines to reviewers as part of the peer review assessment process.

**Qualitative questions.** There was a total of 503 comment responses to the four open-ended questions. Other suggestions on how to improve peer review training included clearer

### What topics would you be most interested in learning?

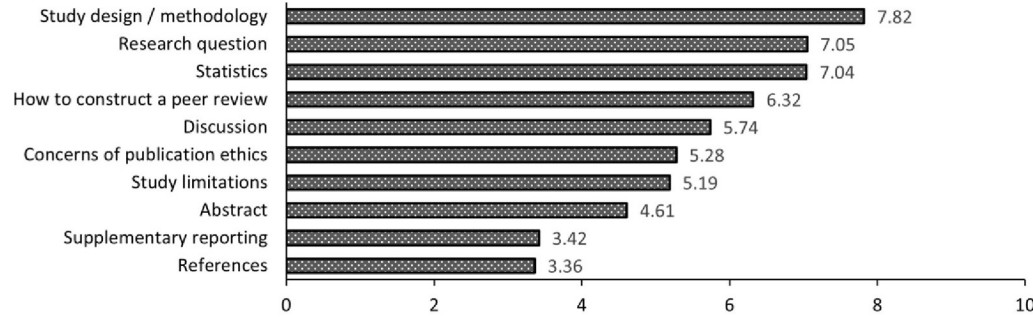

### What organization is best positioned to provide peer review training?

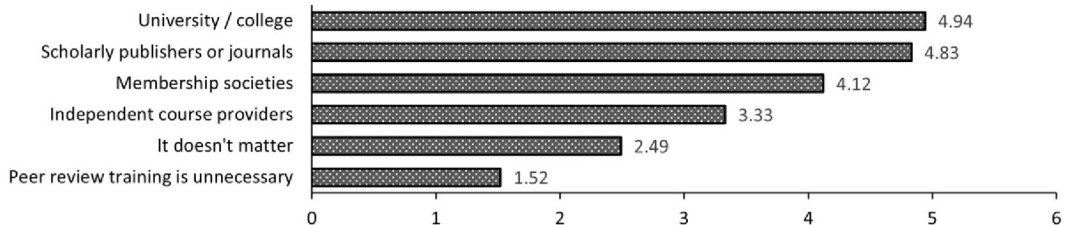

### Who should fund peer review training?

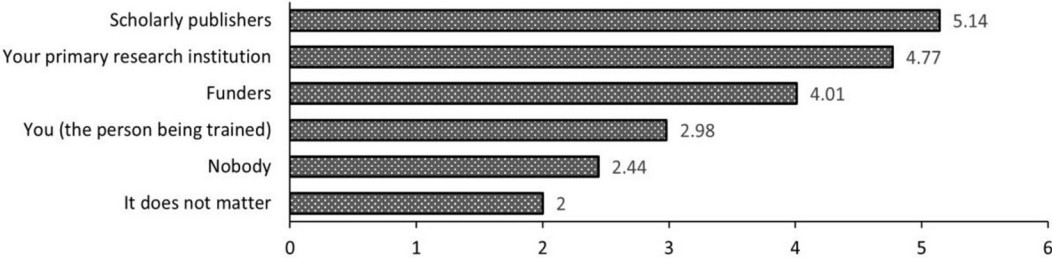

### What would be the best way to deliver peer review training?

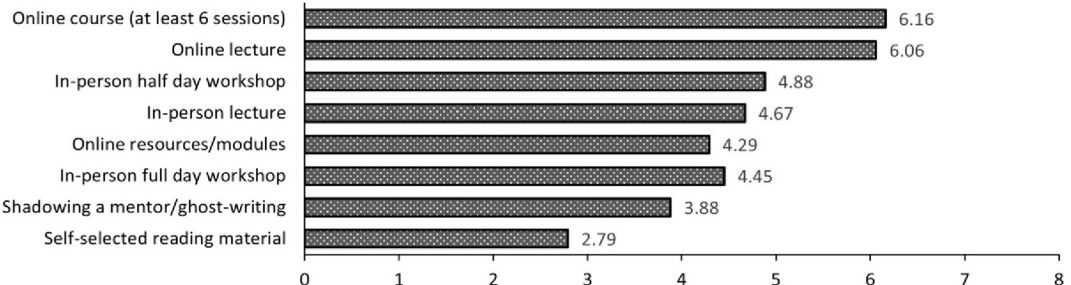

**Fig 2. Ranking of preferred topics, training formats, funding providers, and creating organizations.** Score calculated by average rank placement. A higher score indicates a more preferred and highly ranked response.

standards and expectations (n = 36), improving incentives (n = 32), increased feedback and oversight (n = 28), and improved guidance (n = 19). Barriers to engaging peer review training included difficulty finding or accessing training, including a lack of time or the cost of training offered (n = 111). Desired incentives listed included recognition (n = 39) and financial incentives (n = 35). Please see Table 3 for a list of themes and definitions. In terms of the last

**Table 3. Summary of qualitative survey responses.**

| Item | Number of responses | Themes (examples) | Frequency of themes |
|---|---|---|---|
| Other than training, how do you believe the quality of peer review could be improved? | 152 | **Improve incentives** (official recognition for peer reviewers; monetary incentives; recognition of peer review training for providers; training provided by journals)<br>**Description:** Includes comments that refer to the need to implement more/better incentives for potential reviewers. | 32 |
| | | **Standardization of process** (standardization using templates and/or checklists; standardization through benchmarking; mandatory training certification)<br>**Description:** Includes comments that refer to strategies for standardizing the peer review process to promote structure and consistency. | 11 |
| | | **Clear standards and expectations for peer reviewers** (improved invitation system; focus reviewers toward areas of expertise; provide software for quality checking)<br>**Description:** Includes comments that emphasize the need to focus on selecting appropriate peer reviewers and have clearly defined expectations for said reviewers. | 36 |
| | | **Improved guidance** (mentoring by experienced reviewers; educate reviewers about best practices; provide example reviews from past reviewers; provide written and/or video guidance)<br>**Description:** Includes comments that focus on the importance of providing reviewers with various forms of guidance and/or examples to reference throughout the peer review process. | 19 |
| | | **Feedback and oversight** (better oversight from editors; receiving feedback on peer review comments; collaboration between reviewers during peer review; post-hoc assessment of peer review comments)<br>**Description:** Includes comments that mention the need for improved feedback mechanisms and better oversight from journal editors. | 28 |
| | | **Self-improvement** (personal studies; frequent practice and engagement; individual improvement through increased experience)<br>**Description:** Includes comments that mention the need to encourage reviewers to improve their peer review skills through practice and personal study. | 9 |
| | | **Transparency of peer review** (open peer review; make comments available to all reviewers for comparison; anonymous peer review)<br>**Description:** Includes comments that mention the need for open peer review or the need to maintain reviewer anonymity | 10 |
| What barriers do you face in engaging in peer review training? | 147 | **Complexity/inconsistency of peer review requirements** (differences in specific journal requirements for reviews or training; lack of consensus)<br>**Description:** Includes comments that refer to the lack of clarity/consistency in peer review expectations across journals. | 8 |
| | | **Difficult to find and/or access training** (lack of time; cost of training; availability of training; lack of proper software; internet limitations; lack of international access; unaware of training)<br>**Description:** Includes comments that refer to challenges regarding the availability and/or accessibility of peer review training. For example, lack of access to training may be because of the courses are not offered, or even because busy schedules prevent individuals from completing courses offered at a given time. | 111 |
| | | **Lack of personal incentive** (unwilling attitude; lack of statistics knowledge; lack of recognition; unclear benefit for reviewer; training is monotonous)<br>**Description:** Includes comments that refer to a lack of internal motivation or individual incentives for potential reviewers. | 15 |
| | | **Lack of value placed on peer review/training** (not prioritized by institutes; not sought out; not necessary with expertise)<br>**Description:** Includes comments that refer the lack of value that is placed on peer review training, or peer review in general. This lack of value can either be at the individual or institutional level. | 8 |
| | | **Other** (none)<br>**Description:** Includes other comments that did not clearly fit into one of the previous themes. For example, some respondents reported that they did not face any barriers to peer review training. | 5 |

(*Continued*)

**Table 3.** (Continued)

| Item | Number of responses | Themes (examples) | Frequency of themes |
|---|---|---|---|
| What would incentivize you to obtain additional training in peer review best practices? | 136 | **Financial incentives** (discount for future publications; free training; payment for reviewers; financial support via funding; no monetary incentives ever)<br>**Description:** Includes comments that refer to various forms of financial incentives for potential peer reviewers–either directly related to training or related to peer review more broadly. A minority of respondents also indicated that financial incentives should never be used for peer review. | 35 |
| | | **Recognition of training and/or peer review** (official recognition of training; certification; community commitment; official recognition through research metrics; official recognition from institution; official recognition from journals)<br>**Description:** Includes comments that refer to the need to provide improved institutional/community level recognition to peer reviewers. This can either be recognizing that they completed peer review training, or by recognizing their subsequent peer review work. | 39 |
| | | **Improve feasibility of attending training** (time manageable training; accessible training)<br>**Description:** Includes comments that refer to the need to make peer review training accessible and feasible to accomplish while also attending to other professional/ personal duties. | 11 |
| | | **Translation into the peer review process** (professional requirements for peer review training; high quality training; tangible payoff to training; non-academic benefits for physicians)<br>**Description:** Includes comments that refer to the need for peer review training to be of high quality and clearly beneficial for those who complete it. | 27 |
| | | **Other** (personal interest; none)<br>**Description:** Includes comments that did not clearly fit into one of the previous themes. These comments were from participants who either did not mention any incentives or explicitly stated that no incentives were needed. | 24 |

question which asked for any further comments, responders typically highlighted topics already listed, such as suggestions on how to improve the process (n = 17), suggestions for the implementation of training (n = 16), and the need for incentives (n = 10). The full thematic content analysis can be found in S2 File.

## Discussion

One hundred and eighty-six respondents completed our international survey on peer review training. Among respondents, the vast majority indicated they have never received formal training in peer review. A lack of training could therefore explain the less-than-optimal reporting quality of biomedical research [26] and the inability of reviewers to detect major errors and deficiencies in publications [17].

Limitations of our study included the lower than anticipated response rate. However, this is not out of line with other online surveys, particularly during the COVID-19 pandemic. In addition, most respondents of our study were well established researchers, which was likely a result of our sampling method. Therefore, whether non-responders and early career researchers would respond similarly is unknown.

A majority of respondents either strongly agreed or agreed that peer reviewers should receive formal training. They also indicated that their preference was to receive such training online, as through an online course or lecture. This differs from our recently conducted systematic review which revealed that few online training options exist and of those that do exist, most of the training is a brief one-hour lecture [27]. There appears to be a need to align this disconnect between what peer reviewers want and what is available.

In addition, most respondents indicated difficulty in finding and accessing training in peer review, including not being able to find the time to justify engaging in training. Furthermore, most participants indicated that their primary institution or journal which they worked at did not require/provide training in peer review. Despite this, respondents indicated that scholarly publishers, journals, or universities are best positioned to provide this training. Given academic institutions are training the next generation of researchers, it is surprising that part of a researcher's training does not include such a fundamental part of research. Universities would likely have the expertise and resources to provide such training, given this is often where editors and editorial boards reside. As for journals and scholarly publishers, they may be less inclined to require training given the existing challenges to find peer reviewers. While lowering the barrier is potentially important, this needs to be balanced against the potential risk of providing an unhelpful review and/or missing flaws.

Even if training were available, there may not be enough incentive for peer reviewers, as many respondents in the survey indicated a lack of time or personal benefit. More than a third of respondents reported that recognition for undergoing training or peer reviewing would incentivize them to obtain additional training in peer review. Discussion surrounding incentivizing peer review is part of a broader discourse to move away from the metric of publications to a more societal perspective and reward behaviours that strengthen research integrity [28]. The Declaration of Research Assessment (DORA), the Coalition for Advancing Research Assessment (COARA), and others are now advocating for formally including such activities as part of research(er) assessment [29, 30].

Scholarly peer review plays a crucial role in ensuring the quality of research manuscripts before they are published. Training peer reviewers could enhance reviewer expertise, establish guidelines and standards, and improve review quality. In addition, training could cover important topics such as mitigating bias and publication fraud. We implore stakeholders in peer review to focus future efforts in creating an open and accessible training program in peer review.

## Supporting information

**S1 File. Complete list of survey questions delivered through SurveyMonkey.**
(PDF)

**S2 File. Full thematic content analysis data.**
(XLSX)

**S1 Table. Flowchart of instances of failure of email delivery to intended survey participants.**
(PDF)

**S2 Table. Full survey response data.**
(PDF)

## Author Contributions

**Conceptualization:** Jessie V. Willis, Kelly D. Cobey, David Moher.

**Data curation:** Jessie V. Willis, Janina Ramos.

**Formal analysis:** Jessie V. Willis, Janina Ramos, Hassan Khan, Marc A. Albert, Mohsen Alayche.

**Investigation:** Jessie V. Willis, Janina Ramos, Jeremy Y. Ng.

**Methodology:** Jessie V. Willis, Janina Ramos, Kelly D. Cobey, Jeremy Y. Ng.

**Project administration:** Jessie V. Willis.

**Supervision:** Kelly D. Cobey, David Moher.

**Writing – original draft:** Jessie V. Willis.

**Writing – review & editing:** Jessie V. Willis, Janina Ramos, Kelly D. Cobey, Jeremy Y. Ng, Hassan Khan, Marc A. Albert, Mohsen Alayche, David Moher.

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
