## [Decision Letter · Decision Letter 0]

7 Feb 2023

PONE-D-22-31384Knowledge and motivations of training in peer review: an international cross-sectional surveyPLOS ONE

Dear Dr. Willis,

Thank you for submitting your manuscript to PLOS ONE. After careful consideration, we feel that it has merit but does not fully meet PLOS ONE’s publication criteria as it currently stands. Therefore, we invite you to submit a revised version of the manuscript that addresses the points raised during the review process.

We look forward to receiving your revised manuscript.

Kind regards,

Suhad Daher-Nashif, MSc., PhD

Academic Editor

PLOS ONE

Journal Requirements:

a) Did participants provide their written or verbal informed consent to participate in this study?

Additional Editor Comments:

Dear Dr. Willis,

We've now received the reviewers' comments and recommendations.  

Kindly address each comment in a table, revise your manuscript accordingly with highlighting the changes in red, and re-submit a modified version.

Warmest regards,

Dr Suhad Daher-Nashif

Reviewers' comments:

Reviewer's Responses to Questions

**Comments to the Author**

1. Is the manuscript technically sound, and do the data support the conclusions?

Reviewer #1: Partly

Reviewer #2: Yes

2. Has the statistical analysis been performed appropriately and rigorously? 

Reviewer #1: Yes

Reviewer #2: Yes

3. Have the authors made all data underlying the findings in their manuscript fully available?

Reviewer #1: Yes

Reviewer #2: Yes

4. Is the manuscript presented in an intelligible fashion and written in standard English?

Reviewer #1: Yes

Reviewer #2: Yes

5. Review Comments to the Author

Reviewer #1: Thank you for the opportunity to review this article on the current perceptions of biomedical researchers on peer review. However, whilst it is an interesting read and provides an overview of current perceptions on this topic, in my opinion, there are a number of issues with the article that need addressing, particularly with the discussion. The article feels mismatched in the aims and discussion – with the aim stated as describing an updated perspective on current perceptions of biomedical researchers on peer review; and the discussion reading more as an opinion piece focussing mainly on triangulating findings from this study with a different pre-print review by the same authors as well as recommendations for professionalising peer review. I am not disagreeing with what is being said but it feels that the current findings are being somewhat overstated and I would welcome more interpretation of the survey findings themselves. I have some specific comments for consideration that I believe could improve the manuscript:

Abstract.

Sentence - Most respondents (n = 108, 62.8%) were independent researchers of an academic organization (n = 103, 62.8%) with greater than 21 peer-reviewed articles published (n = 106, 61.6%). This sentence is confusing and it is not clear what the numbers are referring to.

Introduction

Although it is clear why one discipline is being explored, it would be helpful to understand why the field of biomedical research was chosen as this is not explicitly described in the introduction.

Methods.

It would be helpful to know how many potential journals were in the original list, and how many were replaced. Could this be added as a flow chart? Also, it would be helpful to understand why inclusion criteria (e.g. being open access or having access) were not applied to the journal list before randomisation happened (presumably this is due the to the number of journals that there were).

The discussion highlights that the findings from this study match closely to the findings from a pre-print review. Can the authors provided further details on how the survey was developed, including on whether the responses to the questionnaire were informed by the findings of this review

Can authors also reflect on how they mitigated against any potential biases in developing responses to the closed questions or coding of data in the open questions.

Results

Again, this sentence is not clear - The majority (n = 108, 62.8%) were independent researchers defined as assistant, associate or full professors of an academic organization (n = 103, 62.8%) with greater than 21 peer-reviewed articles published (n = 106, 61.6%).

Sentence - For the 27 participants that had received peer review training, … Should this be 26?

Table 2 does not show all of the responses described in the section ‘experience with peer review and peer review training’. It should be made clear that all data can be found in supplemental material and that Table 2 does not present all data.

Figure 2 is very confusing. Some description on how to read this graph would be beneficial.

Sentence Eight (10.0%) indicated that it was required and provided by the journal internally, while two (2.5%) indicated that it was required by externally delivered. Typo on but not by?

The ‘qualitative section’ does not adequately present the open question data. None of themes to arise from the open question are described. A paragraph/summary of the themes would be beneficial, especially if the authors want to only present the top 3 themes by frequency in the table, although I would recommend expanding table 3 to include all of the themes. This table also needs a heading.

The table of themes and frequencies needs further detail explaining that the numbers refer to the frequencies overall and for specific themes.

Discussion

Much of the discussion is given over to mapping a potential path to professionalising peer review. Whilst I agree in principle with much of what is said, it feels out of place here and the findings from the current study being overstated. In particular, the recommendations from paragraph 6 (starting Third…) do not seem to be based on the findings at all.

Furthermore, potential biases and increases in online participation from the pandemic have not been addressed. For example, it is suggested that online training is highly desired. However, the questionnaire responses did not provide the same options for online and face to face and so it might be that lectures and courses are preferred over workshops regardless of whether they are online or face to face.

In addition, the authors cite that a large barrier to receiving training was the limited availability and accessibility of training material. In the context of the other data this reads that there are limited training courses, however, the data show that over 50% of responses were due to the researcher not having the time to engage in these courses. This could perhaps be broken down and expanded on

It is not clear why Appendix 1 is provided. I would recommend this is removed.

Also Supplementary material 2 and 3 show the same information.

Reviewer #2: Well written article. The survey questionnaire addressed pertinent and the analysis is appropriately performed. The discussion is concise and is too the point.

The low response rate has been admitted as a limitation. However, the geographical distribution is biazed naturally towards countries with lots of publications. Perhaps the picture is worse in less developed regions.

6. PLOS authors have the option to publish the peer review history of their article (what does this mean?). If published, this will include your full peer review and any attached files.

Reviewer #1: No

Reviewer #2: No

---

## [Author Response · Author response to Decision Letter 0]

7 Apr 2023

(Please see response to reviewer doc for colour-coded formatting).

Journal Requirements:

This has been done. 

a) Did participants provide their written or verbal informed consent to participate in this study?

This has been done. 

3. Your ethics statement should only appear in the Methods section of your manuscript. If your ethics statement is written in any section besides the Methods, please delete it from any other section. Only appears in Methods section. 

Reviewer #1: Thank you for the opportunity to review this article on the current perceptions of biomedical researchers on peer review. However, whilst it is an interesting read and provides an overview of current perceptions on this topic, in my opinion, there are a number of issues with the article that need addressing, particularly with the discussion. The article feels mismatched in the aims and discussion – with the aim stated as describing an updated perspective on current perceptions of biomedical researchers on peer review; and the discussion reading more as an opinion piece focussing mainly on triangulating findings from this study with a different pre-print review by the same authors as well as recommendations for professionalising peer review. I am not disagreeing with what is being said but it feels that the current findings are being somewhat overstated and I would welcome more interpretation of the survey findings themselves. I have some specific comments for consideration that I believe could improve the manuscript:

We thank the reviewer for their insightful comments. We have provided individual responses below. We have made large changes to the discussion based on these recommendations. 

Abstract.

Sentence - Most respondents (n = 108, 62.8%) were independent researchers of an academic organization (n = 103, 62.8%) with greater than 21 peer-reviewed articles published (n = 106, 61.6%). This sentence is confusing and it is not clear what the numbers are referring to.

This has been changed to “The majority of participants indicated they were men (n = 97 of 170, 57.1%), independent researchers (n = 108 of 172, 62.8%), and primarily affiliated with an academic organization (n = 103 of 170, 62.8%).”

Introduction

Although it is clear why one discipline is being explored, it would be helpful to understand why the field of biomedical research was chosen as this is not explicitly described in the introduction.

We chose biomedical research as this is our content area. The authorship team are in various stages of medical training and/or are faculty members in faculties of medicine. We believe that survey participants (biomedicine) might believe our credibility in this discipline. Similarly, some members of the authorship team have a long history in examining the quality of reporting of biomedical research which is related to the conduct of peer review. We have added a brief explanation to the Introduction section. 

Methods.

It would be helpful to know how many potential journals were in the original list, and how many were replaced. Could this be added as a flow chart? Also, it would be helpful to understand why inclusion criteria (e.g. being open access or having access) were not applied to the journal list before randomisation happened (presumably this is due the to the number of journals that there were).

We have added a breakdown of the reasons for replacement. A total of 26 journals were excluded. We do not believe a flowchart is necessary as there was only a single level of exclusion. There is no way to filter the journals on Scopus by open access/having access via our institution specifically prior to randomization (therefore, it was more feasible to filter them out after the fact). 

No corresponding author name or emails listed (n = 4)

Journal is not in English (n = 7)

No journal website provided from Scopus link/broken link (n = 4)

Journal link dead/inactive after 2020/incorrect (n = 8)

Journal is subscription based and I can't access it via uOttawa (n = 3)

We have added this to the Methods section. 

The discussion highlights that the findings from this study match closely to the findings from a pre-print review. Can the authors provided further details on how the survey was developed, including on whether the responses to the questionnaire were informed by the findings of this review

The survey was not informed directly by the findings of this review as they were developed concurrently. As from the Methods section: The survey was purpose built for this study and administered using SurveyMonkey (https://www.surveymonkey.ca/r/7B2JYR6) software.

Can authors also reflect on how they mitigated against any potential biases in developing responses to the closed questions or coding of data in the open questions.

For coding of data in the open questions, assignment of codes was done independently by two researchers. Following this, individual codes were independently grouped into themes. At both stages, this was finalized through discussion between the two researchers until consensus was reached. This is explained under the “Data Analysis” section of the manuscript. 

We are unsure what is meant by mitigating against potential biases for the closed questions. 

Results

Again, this sentence is not clear - The majority (n = 108, 62.8%) were independent researchers defined as assistant, associate or full professors of an academic organization (n = 103, 62.8%) with greater than 21 peer-reviewed articles published (n = 106, 61.6%).

This has been changed to “The majority of respondents identified as independent researchers defined as assistant, associate, or full professors (n = 108 of 172, 62.8%), were primarily affiliated with an academic organization (n = 103 of 170, 62.8%), and had published more than 21 peer-reviewed articles (n = 106 of 172, 61.6%). “

Sentence - For the 27 participants that had received peer review training, … Should this be 26?

Thank you for noticing this mistake. This has been corrected. 

Table 2 does not show all of the responses described in the section ‘experience with peer review and peer review training’. It should be made clear that all data can be found in supplemental material and that Table 2 does not present all data.

This has been added. 

Figure 2 is very confusing. Some description on how to read this graph would be beneficial.

We have reworked this graph and provided a description. 

Sentence Eight (10.0%) indicated that it was required and provided by the journal internally, while two (2.5%) indicated that it was required by externally delivered. Typo on but not by?

This typo has been corrected. It should have been “but” not “by”. 

The ‘qualitative section’ does not adequately present the open question data. None of themes to arise from the open question are described. A paragraph/summary of the themes would be beneficial, especially if the authors want to only present the top 3 themes by frequency in the table, although I would recommend expanding table 3 to include all the themes. This table also needs a heading.

The table of themes and frequencies needs further detail explaining that the numbers refer to the frequencies overall and for specific themes.

We have significantly expanded on the Qualitative section and reworked the table. Definitions of each theme are provided in the table. 

Discussion

Much of the discussion is given over to mapping a potential path to professionalising peer review. Whilst I agree in principle with much of what is said, it feels out of place here and the findings from the current study being overstated. In particular, the recommendations from paragraph 6 (starting Third…) do not seem to be based on the findings at all.

We have reworked/rewritten the discussion section to address this concern. 

Furthermore, potential biases and increases in online participation from the pandemic have not been addressed. For example, it is suggested that online training is highly desired. However, the questionnaire responses did not provide the same options for online and face to face and so it might be that lectures and courses are preferred over workshops regardless of whether they are online or face to face.

We are unclear on this point. As illustrated in Figure 2, online and in-person options were provided for the question “Which training format would you most prefer?”. 

In addition, the authors cite that a large barrier to receiving training was the limited availability and accessibility of training material. In the context of the other data this reads that there are limited training courses, however, the data show that over 50% of responses were due to the researcher not having the time to engage in these courses. This could perhaps be broken down and expanded on.

We have made the definition of the theme clearer in both the table, Qualitative section, and in the discussion section. We have expanded on this more in both the Qualitative Section and in Discission section. 

It is not clear why Appendix 1 is provided. I would recommend this is removed.

Also Supplementary material 2 and 3 show the same information.

Appendix 1 and Supplementary material 3 have been removed. 

Reviewer #2: Well written article. The survey questionnaire addressed pertinent and the analysis is appropriately performed. The discussion is concise and is too the point.

The low response rate has been admitted as a limitation. However, the geographical distribution is biazed naturally towards countries with lots of publications. Perhaps the picture is worse in less developed regions.

Thank you.

---

## [Decision Letter · Decision Letter 1]

26 Apr 2023

PONE-D-22-31384R1

Knowledge and motivations of training in peer review: an international cross-sectional survey

PLOS ONE

Dear Dr. Willis,

Thank you for submitting your manuscript to PLOS ONE. After careful consideration, we feel that it has merit but does not fully meet PLOS ONE’s publication criteria as it currently stands. Therefore, we invite you to submit a revised version of the manuscript that addresses the points raised during the review process.

We look forward to receiving your revised manuscript.

Kind regards,

Suhad Daher-Nashif, MSc., PhD

Academic Editor

PLOS ONE

Journal Requirements:

Additional Editor Comments:

Thank you for submitting your revised manuscript to PLOS ONE. You addressed most of the comments, but there still a need for minor revisions, in order to be able to make a final decision on your manuscript. 

These comments made by one of the reviewers: 

Figure 2 is now easier to read; however, the description incorrectly identifies the most desired training formats as all online. According to the new figure 2, the online resources/modules were ranked 5th after in person half day workshop and in person lecture.

"The most desired training formats were all online, including online lectures, online courses (at least 6 sessions),

online lectures, and online resources or modules."

The discussion now relates much more to the study findings but ends quite abruptly. I wonder whether a concluding statement would be beneficial; however, I do appreciate that this may be personal preference.

Appendix 2 needs to be renamed Appendix 1; similar for supplemental 4 to 3.

Reviewers' comments:

Reviewer's Responses to Questions

**Comments to the Author**

1. If the authors have adequately addressed your comments raised in a previous round of review and you feel that this manuscript is now acceptable for publication, you may indicate that here to bypass the “Comments to the Author” section, enter your conflict of interest statement in the “Confidential to Editor” section, and submit your "Accept" recommendation.

Reviewer #1: (No Response)

Reviewer #2: All comments have been addressed

2. Is the manuscript technically sound, and do the data support the conclusions?

Reviewer #1: Yes

Reviewer #2: Yes

3. Has the statistical analysis been performed appropriately and rigorously? 

Reviewer #1: Yes

Reviewer #2: Yes

4. Have the authors made all data underlying the findings in their manuscript fully available?

Reviewer #1: Yes

Reviewer #2: Yes

5. Is the manuscript presented in an intelligible fashion and written in standard English?

Reviewer #1: Yes

Reviewer #2: Yes

6. Review Comments to the Author

Reviewer #1: Thank you for responding to all of the comments. A couple of very minor points.

Figure 2 is now easier to read; however, the description incorrectly identifies the most desired training formats as all online. According to the new figure 2, the online resources/modules were ranked 5th after in person half day workshop and in person lecture.

"The most desired training formats were all online, including online lectures, online courses (at least 6 sessions),

online lectures, and online resources or modules."

The discussion now relates much more to the study findings but ends quite abruptly. I wonder whether a concluding statement would be beneficial; however, I do appreciate that this may be personal preference.

Appendix 2 needs to be renamed Appendix 1; similar for supplemental 4 to 3.

Reviewer #2: Thank you for addressing all the responses. I have no concerns about the revised version. I am happy with this version.

7. PLOS authors have the option to publish the peer review history of their article (what does this mean?). If published, this will include your full peer review and any attached files.

Reviewer #1: No

Reviewer #2: No

---

## [Author Response · Author response to Decision Letter 1]

6 Jun 2023

We thank the reviewer for their follow-up comments. We have edited the statement and removed online resources and modules. We have added a concluding paragraph. 

Appendices and supplemental materials have been renamed and removed from text where they do not exist anymore.

---

## [Editor Report · Decision Letter 2]

12 Jun 2023

Knowledge and motivations of training in peer review: an international cross-sectional survey

PONE-D-22-31384R2

Dear Dr. Willis,

We’re pleased to inform you that your manuscript has been judged scientifically suitable for publication and will be formally accepted for publication once it meets all outstanding technical requirements.

Kind regards,

Suhad Daher-Nashif, MSc., PhD

Academic Editor

PLOS ONE

---

## [Editor Report · Acceptance letter]

19 Jun 2023

PONE-D-22-31384R2 

Knowledge and motivations of training in peer review: an international cross-sectional survey 

Dear Dr. Willis:

I'm pleased to inform you that your manuscript has been deemed suitable for publication in PLOS ONE. Congratulations! Your manuscript is now with our production department. 

Kind regards, 

on behalf of

Dr. Suhad Daher-Nashif 

Academic Editor

PLOS ONE